# Experimental Study on the Properties of Simulation Materials for an Aquifuge for a Fluid–Solid Coupling Physical Similarity Model Test

**Xiong Shen [1],\*, Jizu Li [2], Guorui Feng [3], Dekang Zhao [3] and Qin Liu [3]**

[1] College of Security and Emergency Management Engineering, Taiyuan University of Technology, Taiyuan 030024, China

[2] College of Economics and Management, Taiyuan University of Technology, Taiyuan 030024, China; lijizu@tyut.edu.cn

[3] College of Mining Engineering, Taiyuan University of Technology, Taiyuan 030024, China

\* Correspondence: shenxiong0802@163.com; Tel.: +86-159-3413-0429

**Featured Application: Similar materials in aquifuge strata have the characteristics of low strength, controllable water absorption, and low permeability. They were used when testing the fluid–solid coupling physical similarity model in the 8210 working area in Majiliang Coal Mine, which verifies their feasibility and applicability in the research.**

**Abstract:** In order to meet the special requirements of physical and mechanical strength and high water resistance of similar material in aquifuge (aquitard) strata for the testing of the fluid–solid coupling physical similarity model for a mine water inrush. Based on the similarity theory of solid–fluid coupling in equivalent homogeneous continuous media, a new type of aquifuge simulation material was developed, which used river sand as the skeleton of the material, gypsum and calcium carbonate powder as the auxiliary cementing agent, and paraffin wax and petroleum jelly as the waterproof cementing agent. Similar materials of aquifuge (aquitard) strata are created according to a specific proportion of the components and an established technological process. Through orthogonal tests and systematic analysis, the influence mechanism of the different proportions of the raw materials on the variation of the physical–mechanical strength and hydraulic parameters is studied in this paper. The experimental results demonstrate when the mass ratio of solid material to liquid material is 8:1 and 9:1, the mass ratios of river sand, calcium carbonate, and gypsum is 30:3:7, 30:3:7, and 50:3:7, and the mass ratios of paraffin wax to petroleum jelly are 1:2, 1:1, and 2:1, respectively. The controlled ranges of uniaxial compressive strength, softening coefficient, and permeability coefficient of the similar materials are 16.99–426.47 kPa, 0.660–0.805, and $1.01 \times 10^{-7}$– $8.34 \times 10^{-7}$ cm/s, respectively. The above data show that the materials have the characteristics of low strength, controllable water absorption, and low permeability.

**Keywords:** fluid–solid coupling; similar materials modeling; aquifuge; cementing agent; physical–mechanical strength; water-physical parameters

## 1. Introduction

In recent years, the problem of water inrush from coal mines has become increasingly prominent in coal mining. Especially under complex hydrogeological conditions, water fills the goaf. Once the water that fills the goaf reaches a certain water pressure threshold, the water in the goaf breaks through the waterproof rock below the goaf floor and enters the underlying coal face. This situation not only seriously threatens the safety of underground miners, but also causes vast economic losses to the coal mining industry [1]. In view of the engineering problem of water filling in the goaf, at present, there are three research methods that can be applied: theoretical calculation analysis, numerical simulation, and physical model testing. Among them, physical model testing is an important means

to study the deformation and failure of the top and bottom rock layers caused by coal seam mining and the evolution process of the groundwater hydrological cycle. Through physical model testing, the deformation, fracture, and movement of the surrounding rock caused by the excavation of the coal seam can be reproduced, and we can analyze the regularity of the periodic deformation and failure process of the surrounding rock and the characteristics of the groundwater hydrological cycle changes [2]. Based on this, a scientific basis can be provided for the production practice of coal mining engineering. The physical similarity model testing of water filling in the goaf during coal mining considers fluid–solid coupling [3]. In short, "fluid–solid" coupling refers to the coupling effect of the seepage and stress fields. The research on the development and proportion of the similar materials in aquifuge (aquitard) layers is the foundation and guarantee of similarity model testing.

The condition of a "solid–liquid" two-phase medium is the key to simulating the plasticity and hydrologic of aquifuge layers accurately. A considerable number of scholars have conducted research on the fluid–solid coupling of physically similar materials under various material selections and various proportioning schemes. Wang et al. [1] used river sand, nanometer-sized calcium carbonate, gypsum, bentonite, and emulsified wax to prepare similar materials for solid–fluid coupling model testing and found that the structural stability and impermeability are similar to those of sandy mudstone, mudstone, and siltstone, which can be used to study the formation of separated layers of water and the mechanism of water inrush. Sun et al. [4] developed a solid–fluid similar material model to simulate the properties of the surrounding rock in deeply buried mines by using river sand and calcium carbonate as the solid raw materials, paraffin and Vaseline as the cementing agent, and hydraulic oil as regulator, and they applied it to 3D simulation testing of water inrush from a mine floor. Using river sand and clay as the aggregates, machine oil as the modifier, and petroleum jelly as the cementing agent, Zhao et al. [5] developed a type of laterite aquifuge simulation material that can be used to study the crack development law of aquifuge soil layers. Chu et al. [6] studied the low-strength and strong rheological properties using iron ore powder, barite powder, and quartz sand as the aggregates; a rosin alcohol solution as the cementing agent; and hydraulic oil as the adhesive. It could simulate not only the transient elastoplastic properties of soft rock but also similar materials and their rheological characteristics. Li et al. [2] successfully developed a new type of fluid–solid coupling testing of similar material through a large number of proportion tests, using sand, barite powder, and talcum powder as aggregates; cement and Vaseline as the adhesives; and silicone oil as the regulator. It was successfully applied in the model test and research of a Jiaozhou Bay subsea tunnel in Qingdao. Zhao et al. [7] simulated a low strength (0.08–0.56 MPA) solid–fluid coupling similar materials, using fine sand, calcium carbonate, gypsum, paraffin, and Vaseline to study the fracture development of coal and rock mass under mining conditions and the distribution characteristics of water inrush and sand burst channel. Li [8] used sand and talcum powder as the aggregates and paraffin as the adhesive and mixed them with appropriate amounts of the regulator to form solid–fluid coupling similar materials. The paraffin-wax-like materials can solve the problem of solid materials disintegrating in the presence of water. Using stone, quartz sand, and bentonite as the aggregate, and silicon oil and Vaseline as the binders, Huang et al. [3] prepared similar materials with low strength and high plasticity for a water-resisting layer, but the strength was too low and had a general level of plasticity. As it only simulated a weak water barrier, it is difficult to apply these results to harder aquitard layers.

Physical similarity model experiments are important methods for studying specific large-scale engineering laws [9], and many scholars have achieved considerable success in this field of engineering related to seepage and water inrush during the mining process. However, there are still some problems in the fluid–solid coupling similarity model testing of seepage water gushing from mine roof aquifers. First, there are significant differences in the physical strength parameters of similar materials in aquifuge (aquitard) strata under different geometric similarity ratios when applying the fluid–solid coupling similarity simulation theory of underground mining. Another problem is that the physical model

testing of the fluid–structure interaction has high requirements in terms of similar materials. To satisfy the deformation conditions of similar materials in the aquiclude (aquitard), similar conditions of water-physical parameters, such as low permeability, must be met at the same time [10,11]. Furthermore, the physical strength and hydro-chemical stability of similar materials cannot be guaranteed in the development of fluid–solid coupling similar materials, which is mainly manifested in the sudden change in the physical strength and hydro-chemical parameters of seepage water under seepage erosion and hydro-chemical action. Among similar materials, there are often situations where the selected raw materials undergo sustained and uncontrollable hydration reactions with the seepage water, resulting in more uncertainty in the physical model test results. Therefore, it is necessary to conduct further research on similar materials using convection solid coupling.

In this paper, based on the existing research on the fluid–solid coupling of similar materials, river sand was selected as the skeleton of the material, gypsum and calcium carbonate powder as the auxiliary cementing agents, and paraffin wax and petroleum jelly as the waterproof cementing agents of the experimental raw materials. We conduct a comprehensive analysis of physical and mechanical strength and hydraulic parameters to simulate the similar materials of aquifuge (aquitard) strata in the process of fluid–solid coupling physical similarity model testing of "water inrush in goaf exists in the mine". We review the similarity theory of the "solid–liquid" coupling and analyze the mutual regulation mechanism between paraffin wax and petroleum jelly in the physical mechanical strength, plasticity, and waterproofing of the samples. We also study the influence mechanism of physical and mechanical strength changes and the similar change in the hydrologic parameters of the similar material of the aquifuge (aquitard) layers. This study is of great value to the study of water inrush from mined-out areas and to the enrichment and development of fluid–solid coupling physical similarity and model test technology.

## 2. Similarity Theory of Fluid–Solid Coupling

A rock mass is composed of a rock block and a structural plane; in most cases, the rock block also contains small and dense fissures, and the structural plane is composed of fissures, joints, fault strata, unconformity, and foliated surface. When the distribution of these structural planes is relatively uniform, the seepage flow of the rock mass depends on the fissures, and the rock block is relatively impermeable, the rock mass with fissures is regarded as an equivalent continuous medium from a macroscopic point of view. Therefore, the seepage flow of the rock mass can be regarded as an equivalent continuum seepage flow; the permeability coefficient tensor is used to describe the permeability of the rock mass; the stress in the rock mass is regarded as equivalent stress; and the stress tensor is used to describe the stress in the rock mass. Oda [12] and Wu [13] analyzed an equivalent continuous medium model for coupling the seepage and stress fields in rock masses.

According to the similarity principle of physical similarity model testing [14], it is required that the geometric dimensions, boundary conditions, loads, bulk density, strength, deformation characteristics, and hydraulic characteristics of the similar materials of the model follow certain similarity laws. Hu [11] conducted a theoretical analysis of the mathematical model of solid–fluid coupling in homogeneous continuous media. Firstly, the basic mathematical model of the solid–fluid coupling of an equivalent homogeneous continuum was adopted:

(1)  Seepage equation:

$$K_x \frac{\partial^2 p}{\partial^2 x} + K_y \frac{\partial^2 p}{\partial^2 y} + K_z \frac{\partial^2 p}{\partial^2 z} = S \frac{\partial p}{\partial t} + \frac{\partial e}{\partial t} + W \tag{1}$$

(2)  Equilibrium equation:

$$\sigma_{ij,i} + X_j = \rho \frac{\partial^2 u_i}{\partial t^2} \tag{2}$$

(3)    Effective stress equation:

$$\sigma_{ij} = \overline{\sigma_{ij}} + \alpha\delta p \tag{3}$$

The above three equations are the basic solid–fluid coupling equations for homogeneous continuous media, where

$Kx$, $Ky$, and $Kz$ are the permeability coefficients in the three coordinate directions, with $Kx = Ky = Kz$; $P$ is the water pressure; $s$ is the water storage coefficient; $e$ is the volume strain; $W$ is the source-sink term; $\sigma_{ij}$ is the total stress tensor; $\bar{\sigma}_{ij}$ is the effective stress tensor; $X_j$ is the volume force; $\rho$ is the density; $\alpha$ is the effective stress coefficient of Biot; and $\delta$ is the Kronecker sign.

From the equilibrium Equation (2), combined with its geometric equation, 15 basic physical equations are obtained to eliminate the stress and deformation components of the aforementioned equation containing only the displacement component:

$$\begin{cases} G\nabla^2 u + (\lambda + G)\frac{\partial e}{\partial x} + X = \rho\frac{\partial^2 u}{\partial t^2} \\ G\nabla^2 v + (\lambda + G)\frac{\partial e}{\partial y} + Y = \rho\frac{\partial^2 v}{\partial t^2} \\ G\nabla^2 w + (\lambda + G)\frac{\partial e}{\partial z} + Z = \rho\frac{\partial^2 w}{\partial t^2} \end{cases} \tag{4}$$

where $\nabla^2 = \frac{\partial^2}{\partial x^2} + \frac{\partial^2}{\partial y^2} + \frac{\partial^2}{\partial z^2}$ is the Laplace operator symbol. $G = \frac{E}{2(1+\mu)}$ is the shear elastic modulus. $\lambda = \frac{\mu E}{(1+\mu)(1-2\mu)}$ is the Gabriel Lamé constant. $e = \frac{\partial u}{\partial x} + \frac{\partial v}{\partial y} + \frac{\partial w}{\partial z}$ is the volumetric strain. $X$, $Y$, and $Z$ are the volumetric forces in the three coordinate directions.

Let $G' = C_G G''$, $E' = C_E E''$, $x' = C_x x''$, $\lambda' = C_\lambda \lambda''$, $e' = C_e e''$, $u' = C_u u''$, $X' = C_\gamma X''$, $\rho' = C_\rho \rho''$, and $t' = C_t t''$.

Therefore, $\frac{\partial e'}{\partial x'} = \frac{1}{C_l}\frac{\partial e''}{\partial x''}$, $\nabla^2 u' = \frac{C_u}{C_l^2}\nabla^2 u''$, and $\frac{\partial u'}{\partial t'^2} = \frac{C_u}{C_l^2}\frac{\partial^2 u'}{\partial t''^2}$.

Substituting the above relations in the first Equation (4) yields the following:

$$C_G G''\frac{C_u}{C_l^2}\nabla^2 u'' + C_\lambda\lambda''\frac{C_e}{C_l}\frac{\partial e''}{\partial x''} + C_G G''\frac{C_e}{C_l}\frac{\partial e''}{\partial x''} + C_\gamma X'' = C_\rho\rho''\frac{C_u}{C_l^2}\frac{\partial^2 u''}{\partial t''^2} \tag{5}$$

Due to the fact that both the prototype and the model should comply with Equation (4), the following theoretical relationship of the solid–fluid coupling similarity simulation is obtained:

$$C_G\frac{C_u}{C_l^2} = C_\lambda\frac{C_e}{C_l} = C_G\frac{C_e}{C_l} = C_\gamma = C_\rho\frac{C_u}{C_l^2} \tag{6}$$

Formula (6) constitutes a solid–flow coupling equation with a uniform continuous medium. In the formula, $C_G$ is the similarity ratio of the amount of cutting elastic modulus; $C_u$ is the similarity ratio of displacement; $C_l$ is the geometric similarity ratio; $C_\lambda$ is the similarity ratio of the Lamé constant; $C_\gamma$ is the bulk density similarity ratio; $C_e$ is the similarity ratio of volume strain; $C_\rho$ is the density similarity ratio; and $C_t$ is time. $K$ is the penetration coefficient; $p$ is the water pressure; $S$ is the water storage coefficient; $e$ is the volume strain; $W$ is the source of the source; $\rho$ is the density; $\sigma$ is the stress; $G$ is the amount of shear elastic modulus; and $\lambda$ is the Gabriel Lamé's constant.

Suppose that $\Delta_1 = C_G\frac{C_u}{C_l^2}$, $\Delta_2 = C_\lambda\frac{C_e}{C_l}$, $\Delta_3 = C_G\frac{C_e}{C_l}$, $\Delta_4 = C_\gamma$, $\Delta_5 = C_\rho\frac{C_u}{C_l^2}$.

From Formula (6) combined with the theory of similarity, we obtained the following:

① From the similarity of the model ($\Delta_2 = \Delta_3$), we obtained $C_G = C_\lambda$;

② From the similarity of the geometric ($\Delta_1 = \Delta_3$), we obtained $C_u = C_e C_l$. Due to the deformation of the model, its geometry is still similar to that of the prototype, that is, the volume strain similarity ratio, which is $C_u = C_l$;

③ From the similarity of the gravity ($\Delta_3 = \Delta_4$), we obtained $C_G = C_\gamma C_l$;

④ From the similarity of the time ($\Delta_1 = \Delta_5$), we obtained $C_t = \sqrt{C_l}$;

⑤ From the similarity of the stress, we obtained $C_\sigma = C_\gamma C_l$. At the same time, when the equivalent pore pressure coefficient is equal to 1, $C_p = C_\gamma C_l$, that is, the water pressure is similar;

⑥ From the similarity of the source and sink terms, we obtained $C_w = \frac{1}{\sqrt{C_l}}$;

⑦ From the similarity of the storage coefficient, we obtained $C_S = \frac{1}{C_\gamma \sqrt{C_l}}$;

⑧ From the similarity of the permeability coefficient, we obtained $C_K = \frac{\sqrt{C_l}}{C_\gamma}$.

## 3. Development of Fluid–Solid Coupling Similar Materials

### 3.1. Selection of the Similar Materials

Fluid–solid coupling similar materials generally use organic coagulation materials as a binder to ensure the stability of its physical and mechanical strength and to ensure that its percolation water environment does not collapse [15–20]. The geometric similarity ratio of the similarity model is 1:100, and the physics of the similar materials of the aquifuge layers are determined according to the fluid–solid coupling similarity theory of the homogeneous continuum. The requirements of the strength and hydrologic parameters are: the compressive strength range is 0.08–0.5 MPa, the permeability coefficient is $10^{-7}$–$10^{-6}$ cm/s, and the water resistance is high. On the basis of previous studies of the fluid–solid coupling of similar materials, in this paper, paraffin wax and petroleum jelly were chosen as the waterproof cementing agents, river sand as the basic aggregate, and calcium carbonate powder and gypsum were added as auxiliary cementing agents to meet the requirements of the simulation of the seepage of underground water during the mining process. The components of the fluid–solid coupling of similar materials are shown in Figure 1.

(1) Aggregate:

It directly affects certain parameters, such as the bulk density and physical strength of similar materials, and the property stability of the aggregate has a considerable impact on the properties of the similar materials. Therefore, this experiment selected river sand, with relatively stable physical and chemical properties, as the basic aggregate, having a particle size range of 0–2 mm and uniform grading.

(2) Cementing agent:

Its main function is to bond aggregates and to create impermeability. The mechanical strength and hydration strength of the auxiliary cementing agent largely determine the mechanical strength and plastic deformation of the similar materials. The hydrophobicity and flow ability of the waterproof cementing agent comprehensively affect the hydraulic parameters and plastic deformation of the similar materials. Based on the investigation of the properties of the main impermeable and waterproof cementing agents (such as paraffin wax, petroleum jelly, mineral oil, and silicone oil) and auxiliary cementing agents (such as gypsum, calcium carbonate, cement, and graphite), it was found that gypsum is a hygroscopic expansion regulator, calcium carbonate is a good cementing brittleness agent, and petroleum jelly is a chemically inert and hydrophilic waterproof agent, with the characteristics of not easily mixing with water. Solid paraffin wax at a regular temperature is also chemically inert and has excellent non-hydrophilicity and adhesiveness. When it is combined with petroleum jelly to bind river sand aggregate particles, it can disperse water better, and it can easily form a crack joint surface under plastic deformation. The water resistance of similar materials is improved by adding waterproof materials, which can change the crystalline porosity and solubility of the materials and prevent the invasion of water molecules.

In order to meet the performance requirements of similar materials in this simulation experiment, calcium carbonate powder and gypsum were selected as the auxiliary cementing agents, and paraffin wax and petroleum jelly were selected as the main waterproof cementing agents. The characteristics of the selected cementing materials were as follows: No. 58 full-refined paraffin wax; medical-grade white petroleum jelly (Vaseline); calcium

carbonate powder with heavy calcium carbonate and a fineness higher than 1250 mesh; and super-high-quality gypsum powder, with a fineness higher than 350 mesh.

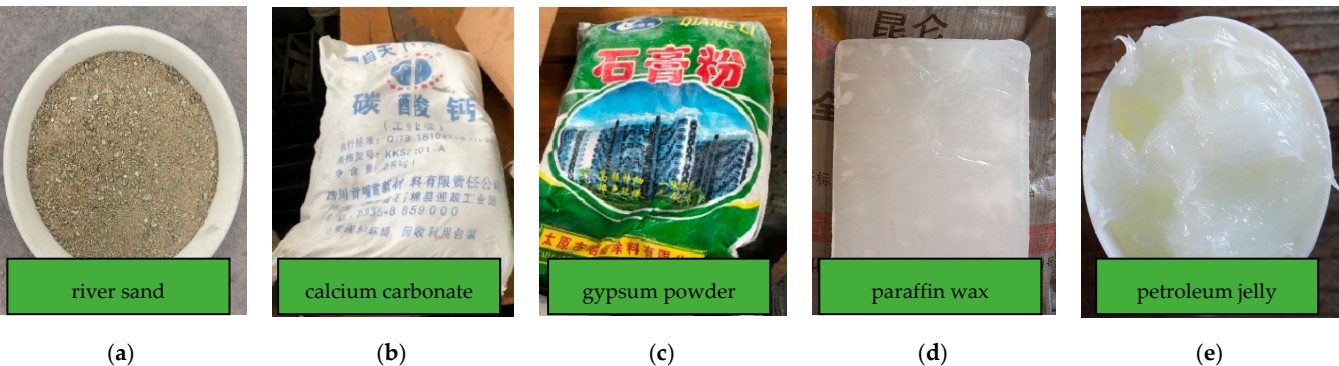

**Figure 1.** The main raw materials to simulate the similar materials for the water-resistant layer: (**a**) river sand; (**b**) calcium carbonate; (**c**) gypsum; (**d**) paraffin wax; and (**e**) petroleum jelly.

*3.2. Specimen Preparation*

The manufacture of similar materials containing solid paraffin wax requires heating, and the equipment for making the test pieces must be kept at a specific temperature before the test materials are molded and hardened. Considering that the mixed materials after uniform mixing are prone to adhesion to the mold during the pressing process of the test piece, and in order to ensure that the test piece has a smooth and complete surface after demolding, a double-cast-iron mold with evenly applied lubricating oil on the internal surface of the mold was used to make a cylindrical sample. The sample size was $\varphi = 50$ mm $\times$ 100 mm, which meets the requirements of the uniaxial compression of the specimen.

In light of the composition and properties of the similar materials for the aquiclude, the preparation process is divided into the following steps:

Firstly, the aggregate and the auxiliary cement are stirred evenly.

Weigh the corresponding amounts of river sand, calcium carbonate powder, and gypsum according to the predetermined distribution ratio of the similar materials in each group. Place the mixed aggregate and auxiliary cementing agent in a stirring pot, at a constant temperature, and stir uniformly according to the dry method.

Next, the waterproof cementing agent is mixed evenly.

Weigh the mixture of impermeable waterproof cementing agent, paraffin wax and petroleum jelly and place in a heating device at the constant temperature of 60 °C; heat it to melt, and stir continuously and evenly. The heating temperature should not be too high or too low. If the temperature is too high, the paraffin ignites directly. If the temperature is too low, the mixed materials do not melt quickly and are not fully mixed. Therefore, a 60 °C thermostat was used to heat the mixture consisting of solid paraffin wax and petroleum jelly at room temperature until it was completely liquid.

Then, the components of the similar materials are mixed and stirred evenly.

Slowly inject the mixed waterproof cementing agent liquid into the uniform mixture of the aggregate and auxiliary cementing agent in the stirring pot at a constant temperature of 60 °C. Keep stirring until the waterproof cementing agent and aggregate and auxiliary cement are mixed evenly.

Next, the similar materials after mixing are molded and pressed.

If the temperature of the mixed material decreases too quickly, the properties of the test materials will be affected. Therefore, it is necessary to keep the temperature of the stirring pot in the range of 50–60 °C during the molding process of similar materials. To ensure successful specimen pressing and demolding, it is necessary to apply lubricating oil to the inner surface of the mold before making the waterproof specimens. Pour the mixture into the specimen mold quickly, layer by layer, and compact it to complete the molding and pressing process.

Finally, the similar material samples are cooled and molded.

Due to its expected low physical and mechanical strength, the sample of similar material is demolded after cooling to room temperature and cured at room temperature for 7 days to complete drying and hardening.

The simulation specimens made of similar materials are shown in Figure 2.

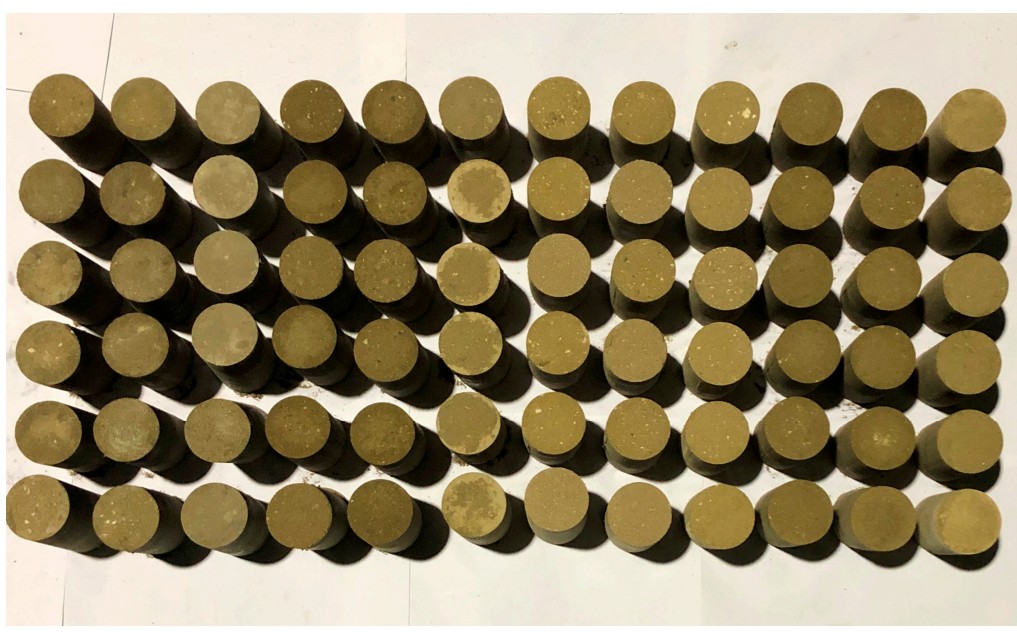

**Figure 2.** The specimens made of aquiclude similar materials.

*3.3. Testing of the Basic Mechanical Parameters of Similar Materials*

According to the requirements of the "Standard for Engineering Rock Mass Testing Methods" (GB/T50266-2013), the testing of the basic mechanical parameters of similar materials includes determining the uniaxial compressive strength of the materials. Based on the engineering geological background of the Majiliang Coal Mine, the experimental bench specifications, and similarity theory, it was determined that the required compressive strength of the sample to be made should be around 0.085–0.252 MPa. The ratio of the bonding materials was adjusted to produce multiple sets of samples with different ratios.

The test of compressive (tensile) strength was conducted on a TAW-2000 electro-hydraulic servo-rock rigid pressure testing machine (Changchun Chaoyang test instrument). Three specimens were tested in each ratio group, and the average value was taken as the final mechanical strength value under the conditions of each ratio. Considering that the mechanical strength of the tested similar materials is mainly affected by the aggregate ratio of river sand, calcium carbonate powder, and gypsum, and the weight ratio of paraffin wax to petroleum jelly in the mixture, an orthogonal test method was used to determine the distribution ratio of each group [21], and the uniaxial compressive strength of the similar materials was obtained.

The results of the uniaxial compressive strength testing are shown in Table 1.

The test results show that the compressive strength of the material has a wide adjustment range, and the range of uniaxial compression strength is 16.99~426.47 kPa, which can be used to simulate different types of rock with medium and low strength. Among the components of similar materials, the change in the contents of the cementing agent has a very evident impact on the overall strength of the materials. Adjusting the content of paraffin wax and petroleum jelly has a significant impact on the compressive strength of the materials. Figure 3 shows the influence curve of the solid–liquid materials' mass ratios on the compressive strength of the materials with different paraffin wax and petroleum jelly mass ratios.



**Table 1.** Results of the uniaxial compression tensile strength testing.

| Number | Ratio of Solid–Liquid Components | Ratio of Paraffin Wax–Petroleum Jelly | Ratio of Sand–Calcium Carbonate–Gypsum | Uniaxial Compressive Strength /kPa |
|---|---|---|---|---|
| A-1 | 7:01 | 1:01 | 20:03:07 | 364.36 |
| A-2 | 7:01 | 1:01 | 30:03:07 | 350.34 |
| A-3 | 7:01 | 1:01 | 40:03:07 | 310.28 |
| A-4 | 7:01 | 1:01 | 50:03:07 | 294.36 |
| A-5 | 7:01 | 1:01 | 60:03:07 | 288.82 |
| A-6 | 7:01 | 1:02 | 20:03:07 | 334.81 |
| A-7 | 7:01 | 1:02 | 30:03:07 | 305.28 |
| A-8 | 7:01 | 1:02 | 40:03:07 | 299.9 |
| A-9 | 7:01 | 1:02 | 50:03:07 | 270.12 |
| A-10 | 7:01 | 1:02 | 60:03:07 | 229.96 |
| A-11 | 7:01 | 2:01 | 20:03:07 | 426.47 |
| A-12 | 7:01 | 2:01 | 30:03:07 | 418.26 |
| A-13 | 7:01 | 2:01 | 40:03:07 | 370.79 |
| A-14 | 7:01 | 2:01 | 50:03:07 | 323.24 |
| A-15 | 7:01 | 2:01 | 60:03:07 | 275.69 |
| B-1 | 8:01 | 1:01 | 20:03:07 | 325.64 |
| B-2 | 8:01 | 1:01 | 30:03:07 | 303.45 |
| B-3 | 8:01 | 1:01 | 40:03:07 | 286.53 |
| B-4 | 8:01 | 1:01 | 50:03:07 | 259.08 |
| B-5 | 8:01 | 1:01 | 60:03:07 | 234.4 |
| B-6 | 8:01 | 1:02 | 20:03:07 | 287.06 |
| B-7 | 8:01 | 1:02 | 30:03:07 | 250.01 |
| B-8 | 8:01 | 1:02 | 40:03:07 | 211.74 |
| B-9 | 8:01 | 1:02 | 50:03:07 | 175.92 |
| B-10 | 8:01 | 1:02 | 60:03:07 | 138.87 |
| B-11 | 8:01 | 2:01 | 20:03:07 | 390.23 |
| B-12 | 8:01 | 2:01 | 30:03:07 | 351.27 |
| B-13 | 8:01 | 2:01 | 40:03:07 | 310.85 |
| B-14 | 8:01 | 2:01 | 50:03:07 | 273.36 |
| B-15 | 8:01 | 2:01 | 60:03:07 | 259.08 |
| C-1 | 9:01 | 1:01 | 20:03:07 | 135.91 |
| C-2 | 9:01 | 1:01 | 30:03:07 | 114.41 |
| C-3 | 9:01 | 1:01 | 40:03:07 | 88.76 |
| C-4 | 9:01 | 1:01 | 50:03:07 | 62 |
| C-5 | 9:01 | 1:01 | 60:03:07 | 35.79 |
| C-6 | 9:01 | 1:02 | 20:03:07 | 79.42 |
| C-7 | 9:01 | 1:02 | 30:03:07 | 63.81 |
| C-8 | 9:01 | 1:02 | 40:03:07 | 48.73 |
| C-9 | 9:01 | 1:02 | 50:03:07 | 32.6 |
| C-10 | 9:01 | 1:02 | 60:03:07 | 16.99 |
| C-11 | 9:01 | 2:01 | 20:03:07 | 157.15 |
| C-12 | 9:01 | 2:01 | 30:03:07 | 125.43 |
| C-13 | 9:01 | 2:01 | 40:03:07 | 92.85 |
| C-14 | 9:01 | 2:01 | 50:03:07 | 71.42 |
| C-15 | 9:01 | 2:01 | 60:03:07 | 44.42 |

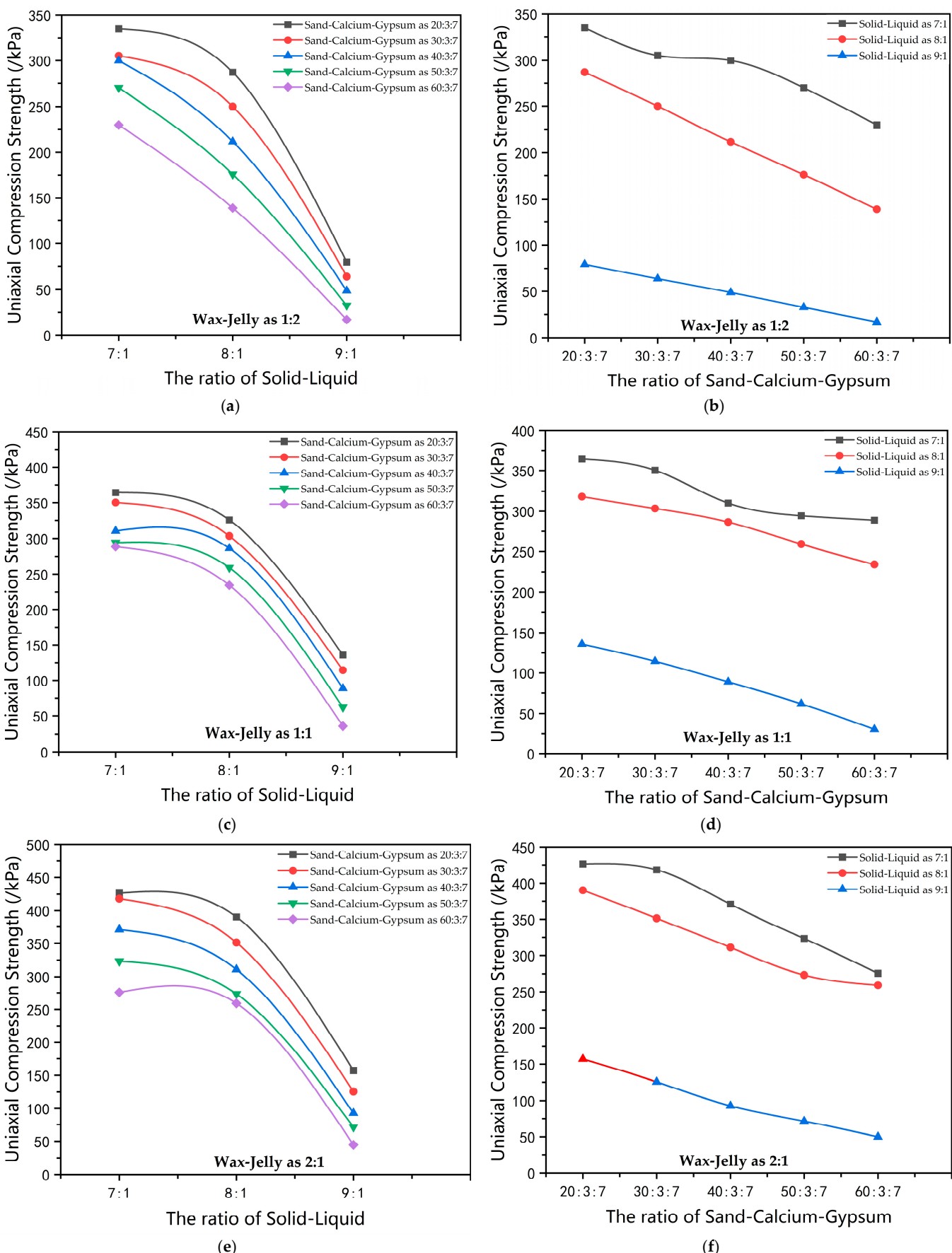

**Figure 3.** The influence curve of the mass ratio of solid–liquid materials on the compressive strength of the material. (**a**) Influence curve of the ratio of solid–liquid materials on the uniaxial compression

strength when the paraffin wax–petroleum jelly ratio is 1:2. (**b**) Influence curve of the ratio of river sand–calcium carbonate–gypsum on the uniaxial compression strength when the paraffin wax–petroleum jelly ratio is 1:2. (**c**) Influence curve of the ratio of solid–liquid materials on the uniaxial compression strength when the paraffin wax–petroleum jelly ratio is 1:1. (**d**) Influence curve of the ratio of river sand–calcium carbonate–gypsum on the uniaxial compression strength when the paraffin wax–petroleum jelly ratio is 1:1. (**e**) Influence curve of the ratio of solid–liquid materials on the uniaxial compression strength when the paraffin wax–petroleum jelly ratio is 2:1. (**f**) Influence curve of the ratio of river sand–calcium carbonate–gypsum on the uniaxial compression strength when the paraffin wax–petroleum jelly ratio is 2:1.

From Figure 3, it can be observed that, when the mass ratio of river sand–calcium carbonate–gypsum and the mass ratio of paraffin wax–petroleum jelly are fixed, the uniaxial compressive strength of the specimen decreases with the increase in the mass ratio of solid to liquid raw materials, which indicates that the mass ratio of solid–liquid raw materials is a decisive factor in the adjustment of the uniaxial compressive strength of the specimen. When the river sand–calcium carbonate–gypsum ratio is 20:3:7 and the paraffin wax–petroleum jelly ratio is 1:2, the compressive strength of the specimen is 334.81 kPa when the ratio of solid–liquid materials is 7:1. When the ratio of solid–liquid materials is 8:1, the compressive strength of the specimen is 287.06 kPa, which is reduced by 14.26% compared with the previously mentioned ratio of the solid–liquid materials. When the ratio of solid–liquid materials is 9:1, the compressive strength of the specimen is 79.42 kPa, which is reduced by 76.28% compared with the 7:1 ratio of solid–liquid materials. The results show that the strength of similar materials is negatively correlated with the ratio of solid–liquid materials, when the ratios of paraffin wax–petroleum jelly were 1:2 and 2:1; when the ratios of solid–liquid materials were 7:1 and 8:1, the effect of the proportion of river sand content on the strength of the similar materials was larger than that of the similar materials with a ratio of solid–liquid materials of 9:1. When the ratio of paraffin wax–petroleum jelly was 1:1, the strength of the material changes more uniformly with the change in the ratio of solid–liquid materials and the proportion of river sand content.

From Figure 4, it can be observed that, under a certain solid–liquid ratio, as the paraffin wax–petroleum jelly ratio increases from 1:2 to 2:1, the compressive strength of the similar materials gradually increases with the increase in the content of auxiliary cementing agent. When the content of river sand in the liquid–solid raw materials is lower than 75%, the strength of the similar materials increases slowly with the increase in the auxiliary cementing agent. The strength of the material at this point is mainly provided by the bond of the waterproof cementing agent. When the content of river sand in the solid raw materials exceeds 75%, the strength of the similar material is mainly controlled by the proportion of the content of the auxiliary cementing agent, and the strength of the similar material increases linearly with the increase in the content of gypsum and calcium carbonate powder.

According to the analysis of the influence curve of the uniaxial compression strength of the similar materials in the figures above, it can be concluded that, in the uniaxial compressive strength of similar materials, of the four cementing agents, the hydration–hardening cementation process of gypsum is similar to that of solid paraffin wax at room temperature, which changes mainly with the content amount. Calcium carbonate powder does not react with water but plays a role in filling gaps in similar materials. Petroleum jelly as a waterproof cementing agent is a type of plastic binder. When the content of other components is fixed, the strength of similar materials decreases with the increase in petroleum jelly content.

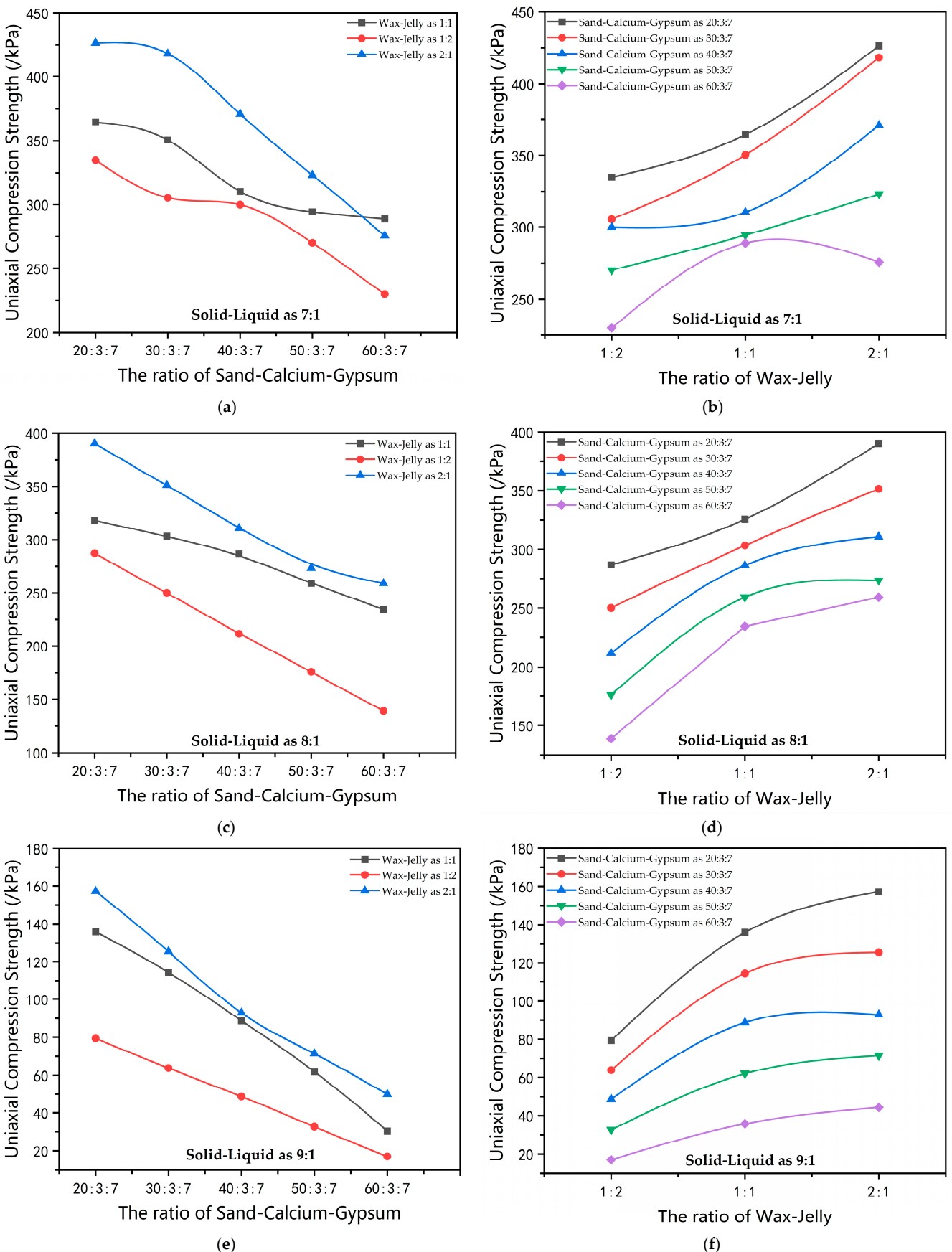

**Figure 4.** The influence curve of the ratio of solid–liquid materials on the compressive strength of the material. (**a**) Influence curve of the ratio of river sand–calcium carbonate–gypsum on the

uniaxial compression strength when the solid–liquid materials ratio is 7:1. (**b**) Influence curve of the ratio of paraffin wax–petroleum jelly on the uniaxial compression strength when the solid–liquid materials ratio is 7:1. (**c**) Influence curve of the ratio of river sand–calcium carbonate–gypsum on the uniaxial compression strength when the solid–liquid materials ratio is 8:1. (**d**) Influence curve of the ratio of paraffin wax–petroleum jelly on the uniaxial compression strength when the solid–liquid materials ratio is 8:1. (**e**) Influence curve of the ratio of river sand–calcium carbonate–gypsum on the uniaxial compression strength when the solid–liquid materials ratio is 9:1. (**f**) Influence curve of the ratio of paraffin wax–petroleum jelly on the uniaxial compression strength when the solid–liquid materials ratio is 9:1.

The physical and mechanical strength of the similar materials comes mainly from the physico-chemistry adhesion of the binder, and the strength of the aggregate itself has a great influence on it. The paired use of calcium carbonate powder and gypsum as auxiliary binders, paraffin wax and petroleum jelly as waterproof binders is an application principle of binary mutual adjustment. The objective physical and mechanical strength of the similar materials and the comprehensive application of water physics can be achieved by the reasonable proportion of mutually adjusting the binary elements in the binder.

*3.4. Water Resistance Experiment and Analysis of the Results*

The goals of this experiment were two-fold: the similar materials must have a certain physical and mechanical strength, and the similar material of this aquifuge (aquitard) layer has a very weak permeability and water absorption when the fluid–solid coupling model is tested. The similar materials should maintain a certain strength without loosening under the conditions of seepage and water immersion. Therefore, compressive strength, water immersion, permeability, and water absorption tests were conducted. Some of the experimental scheme and parameter test results are shown in Table 2.

**Table 2.** The water-physical parameters of the typical contents of the tested blocks.

| Number | Ratio of Solid–Liquid Components | Ratio of Paraffin Wax–Petroleum Jelly | Ratio of River Sand–Calcium Carbonate–Gypsum | Softening Coefficient | Water Absorption % | Permeability Coefficient $10^{-7}$ cm/s |
|---|---|---|---|---|---|---|
| B-2 | 8:01 | 1:01 | 30:03:07 | 0.783 | 7.78 | 1.16 |
| B-3 | 8:01 | 1:01 | 40:03:07 | 0.78 | 6.86 | 1.53 |
| B-4 | 8:01 | 1:01 | 50:03:07 | 0.759 | 6.09 | 1.87 |
| B-7 | 8:01 | 1:02 | 30:03:07 | 0.739 | 7.51 | 4.08 |
| B-8 | 8:01 | 1:02 | 40:03:07 | 0.72 | 7.12 | 5.36 |
| B-9 | 8:01 | 1:02 | 50:03:07 | 0.699 | 5.8 | 7.87 |
| B-12 | 8:01 | 2:01 | 30:03:07 | 0.805 | 3.13 | 1.01 |
| B-13 | 8:01 | 2:01 | 40:03:07 | 0.68 | 2.87 | 1.46 |
| B-14 | 8:01 | 2:01 | 50:03:07 | 0.448 | 2.43 | 1.81 |
| C-2 | 9:01 | 1:01 | 30:03:07 | 0.781 | 9.78 | 3.89 |
| C-3 | 9:01 | 1:01 | 40:03:07 | 0.73 | 8.65 | 4.73 |
| C-4 | 9:01 | 1:01 | 50:03:07 | 0.686 | 7.83 | 6.72 |
| C-7 | 9:01 | 1:02 | 30:03:07 | 0.677 | 11.45 | 4.37 |
| C-8 | 9:01 | 1:02 | 40:03:07 | 0.66 | 10.19 | 5.83 |
| C-9 | 9:01 | 1:02 | 50:03:07 | 0.664 | 8.25 | 8.34 |
| C-12 | 9:01 | 2:01 | 30:03:07 | 0.758 | 5.17 | 1.18 |
| C-13 | 9:01 | 2:01 | 40:03:07 | 0.77 | 4.67 | 1.65 |
| C-14 | 9:01 | 2:01 | 50:03:07 | 0.758 | 2.37 | 1.96 |

3.4.1. Water Immersion Test

In the water immersion test, when the amount of paraffin wax in the sample was 0, we placed it in water and it started to dissolve immediately, so it was not necessary to study. Under a certain mass ratio of river sand, calcium carbonate powder, and gypsum, the added amounts of paraffin wax were 3.33%, 3.70%, 5.00%, 5.56%, 6.67%, and 7.41%. The samples were weighed in water for 24 h to obtain 24 h water absorption values, and

then the softening coefficient of similar materials was obtained by uniaxial compressive strength and degree test.

(1) Water Absorption

The water absorption characteristics of similar materials are mainly characterized by water absorption parameters; the lower the water absorption, the stronger the non-hydrophilicity [10]. The formula for calculating water absorption is as follows:

$$a = (m_w / m_d) \times 100\% \tag{7}$$

where $a$ is the water absorption; $m_w$ is the water absorption of the specimen after 24-h immersion; and the increment in the specimen's mass before and after immersion was calculated. $m_d$ is the mass of the sample before soaking.

According to the experimental data, the influence curves of the amount of paraffin on the 24hour water absorption (24H water absorption) and softening the coefficient of the sample were drawn and shown in Figures 5 and 6, respectively.

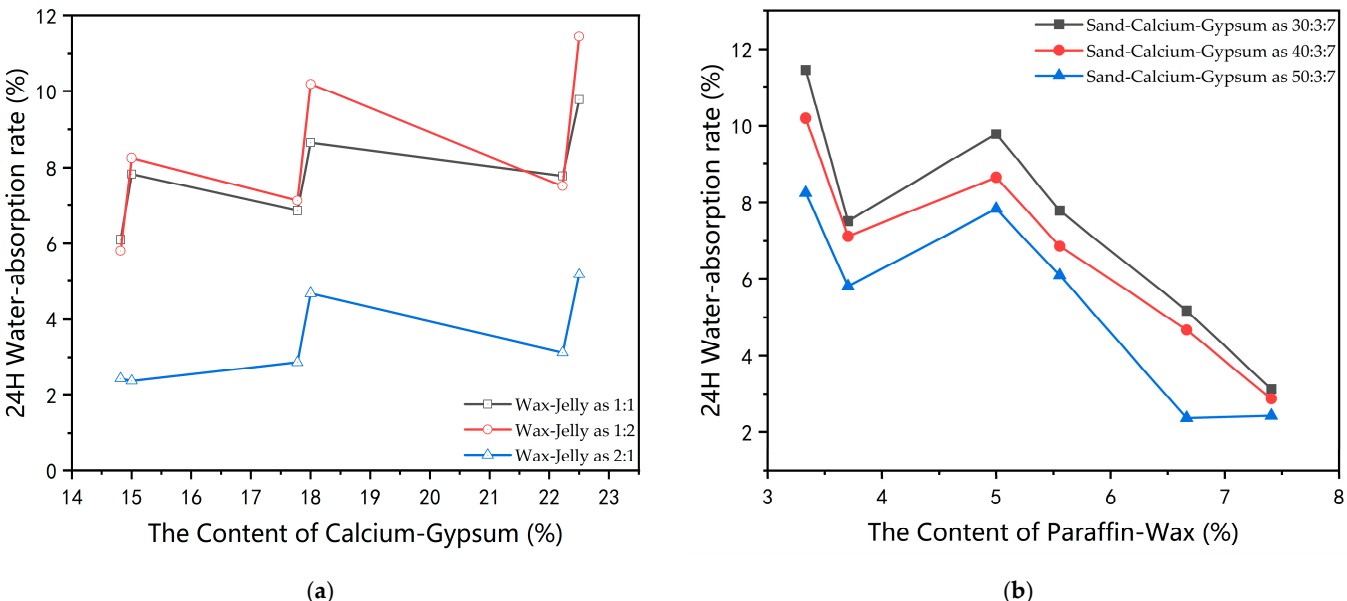

(a)                                                                 (b)

**Figure 5.** The influence curve of the content of paraffin wax and the percentage of the auxiliary cementing agent on the water absorption of the similar materials. (**a**) The influence curve of the content of the auxiliary cementing agent on the water absorption. (**b**) The influence curve of the content of paraffin wax on the water absorption.

According to the comprehensive analysis shown in Figure 5a,b, river sand is the main aggregate under different ratios of solid materials, mainly constituting the pore volume of the sample. At the same time, the main pore of the sample through which water absorption occurs is filled by calcium carbonate and gypsum, that is, the value of 24H water absorption does not change significantly.

When the proportion of solid materials is fixed, the paraffin wax content of the cementing agent increases, and the 24H water absorption is mainly related to the pore volume of the standard sample. From Figure 5b, it can be seen that the effective pore volume of water absorption decreases with the increase in paraffin wax content. The mixture of molten paraffin wax and petroleum jelly was used in binding the similar aggregate materials of the water-resisting layer, and paraffin was the main sealing component of the water-absorbing pore of the sample, so the value of 24H water absorption rate decreased gradually. Therefore, in Figure 5b, with a river sand–calcium carbonate–gypsum ratio of 40:3:7, as the paraffin wax content increases from 3.33% to 7.41%, the value of 24H water absorption of the similar materials sample generally decreases from 10.19 to 2.87. Among

them, when the paraffin wax content accounts for 5%, the 24H water absorption value suddenly increases to 8.65.

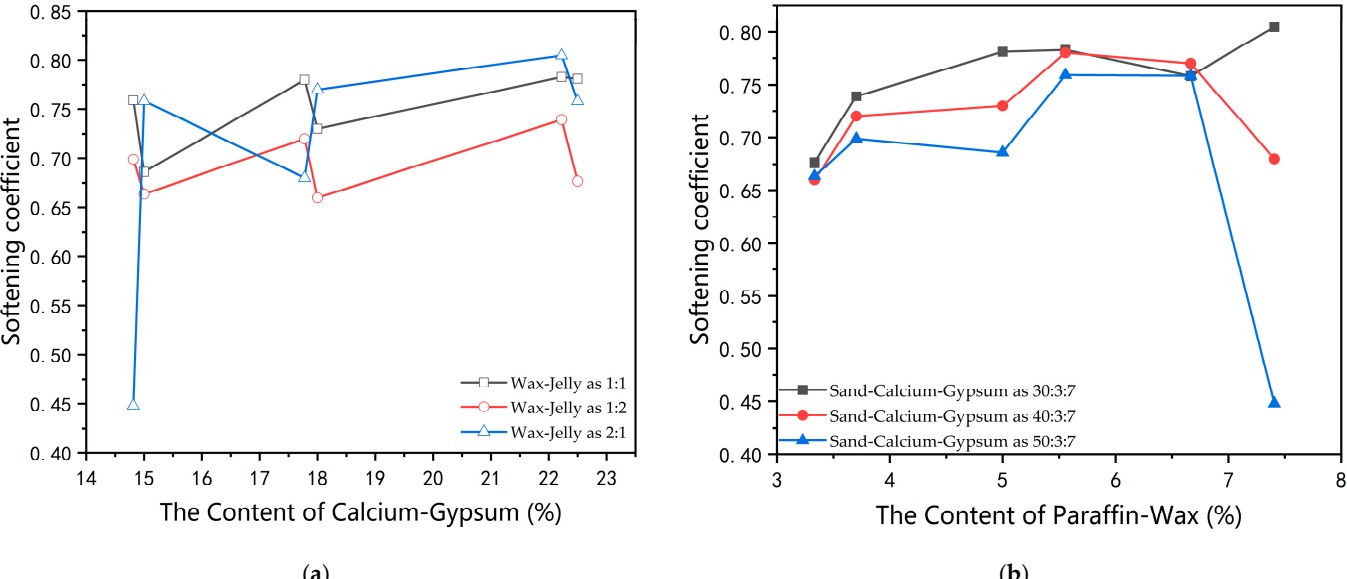

**Figure 6.** The influence curve of the content of paraffin wax and the percentage of auxiliary cementing agent on the softening coefficient of the material. (**a**) The influence curve of the percentage of calcium carbonate and gypsum on the softening coefficient. (**b**) The influence curve of the content of paraffin wax on the softening coefficient.

(2)　Softening coefficient

　　Among the hydraulic properties of similar materials, the softening coefficient is an important comprehensive indicator of changes in water ratio and strength of the materials. It mainly characterizes the decrease in strength and increase in flexibility (decrease in elastic modulus) of similar materials after being saturated with water. It is called the softening property and is represented by the softening coefficient. $K_R$ is defined as the saturated compressive strength of rock specimens ($\sigma_{cw}$) and dry compressive strength ($\sigma_c$). The calculation formula is as follows:

$$K_R = \sigma_{cw} / \sigma_c \qquad (8)$$

where $K_R$ is the softening coefficient of the similar materials, $\sigma_{cw}$ is the saturated compressive strength, and $\sigma_c$ is the dry compressive strength. The test results of the softening coefficient are shown in Table 2 and Figure 6.

　　The softening coefficient parameter is directly related to the uniaxial compressive strength of the specimen after being saturated with water. When the ratio of solid materials is fixed, and with the increase in paraffin wax content in the binder, the 24H water absorption decreased gradually. That is, the water content of the sample decreased after water absorption. Therefore, the weakening effect of water on the strength of the sample also shows a decreasing trend. From Figure 6b, it can be observed that, when the proportion of paraffin content is 3.33–6.67%, the softening coefficient increases slightly with the increase in paraffin content in similar materials. When the river sand–calcium carbonate–gypsum ratio is 30:3:7, the softening coefficient of the sample increases slowly from 0.677 to 0.758 with the increase in paraffin wax content from 3.33% to 6.67%.

　　In addition, from Figure 6a,b, we can observe that the softening coefficients of the samples, for ratios of river sand–calcium carbonate–gypsum of 30:3:7, 40:3:7, and 50:3:7, were very close for paraffin contents of 3.33% and 6.67%, respectively. When the proportion of paraffin is between 3.33% and 6.67%, the softening coefficient increases with the increase in the proportion of the auxiliary cementing agent; at the same time, the water absorption strength of the sample is stronger. This is because the samples with the same proportion

of paraffin wax had the same amount of water absorption, but calcium carbonate and gypsum play a role in the auxiliary cementation process, so that the specimen is further enhanced after the strength added by water absorption. However, when the proportion of paraffin wax was 7.41%, the softening coefficient of the samples with the three ratios of river sand–calcium carbonate–gypsum was reversed. The river sand–calcium carbonate–gypsum ratio was 50:3:7, and the softening coefficient of the samples with the proportion of calcium carbonate–gypsum was 14.84%, which suddenly dropped to 0.448. When the amount of river sand is too large, the content of calcium carbonate–gypsum that helps in the cementation process is not enough to complete the hydration–cementation process of a large number of aggregates. The increase in the paraffin wax content reduces the water absorption of the sample, and the inclusion of the auxiliary cementing agent cannot be completed, so this situation occurs.

Through comprehensive analysis, it can be observed that, from the softening coefficient change of the similar material after immersion, the strength of the model sample after immersion is slightly reduced compared to that without immersion, with a reduction range of 0.35–0.2. The influence of water absorption on the strength of similar materials of water-resisting layers is a rather complex process.

### 3.4.2. Permeability Test

Under the action of a specific hydraulic gradient, the property that similar materials display when permeated by water is called permeability. The permeability coefficient is an important characteristic indicator that characterizes the permeability of similar materials. It is generally considered that the flow of water in similar materials, as in the flow of water in porous media, such as soil, is also subject to the linear seepage law, i.e., Darcy's law, expressed as

$$U = KJ \tag{9}$$

where $U$ is the seepage velocity, $J$ is the hydraulic gradient, and $K$ is the permeability coefficient, which is numerically equal to the seepage velocity when the hydraulic gradient is 1.

The conventional laboratory permeability test can be divided into two types. The constant head permeability test is normally used for a material with a large permeability ($K > 10^{-3}$ cm/s), while the variable head permeability test is used for a composite material with a low permeability ($K < 10^{-3}$ cm/s). In this paper, the impermeability characteristics of the grout were characterized by two measured parameters, that is, the impermeability coefficient and osmotic pressure. The permeability coefficient was measured by a TST-55 model permeameter [22]. The variable head permeability coefficient was calculated as follows:

$$K_{\mathrm{T}} = 2.3 \frac{aL}{A(t_2 - t_1)} \log \frac{H_1}{H_2} \tag{10}$$

where $A$ is the cross-sectional area of the head pipe (cm$^2$); $a$ is the sample cross-sectional area (cm$^2$); 2.3 is the conversion factor of ln and log; $L$ is the permeability diameter, that is, the height of the sample (cm); $t_1$ and $t_2$ are the start and end times (s) of the head measurement, respectively; and $H_1$ and $H_2$ are the start and end of the water head, respectively.

The permeability coefficient of the similar materials was measured by the variable head permeability test method. The test data are shown in Table 2 and Figure 7.

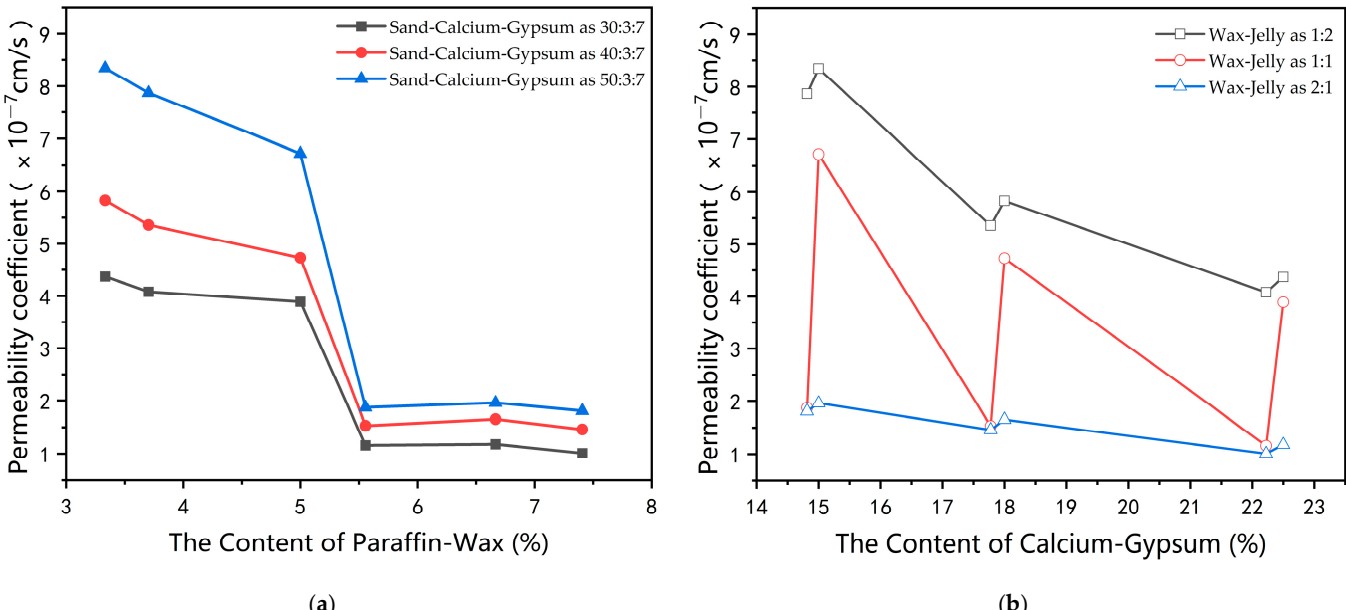

**Figure 7.** The influence curve of the content of paraffin and the percentage of calcium carbonate–gypsum on the permeability coefficient of the material. (**a**) The influence curve of the content of paraffin on the permeability coefficient. (**b**) The influence curve of the percentage of calcium carbonate–gypsum on the permeability coefficient.

Figure 7a shows the effect of paraffin wax content on the permeability coefficient of the similar materials. The proportion of paraffin wax content increases under a certain ratio of river sand–calcium carbonate–gypsum. The gaps between the particles of river sand, which is the main aggregate of similar materials, are evenly mixed and filled with the cementing agent. The permeability coefficient of similar materials decreases with the increase in water resistance and the decrease in permeability space. Under a certain mass ratio of paraffin wax, with the increase in the proportion of river sand, the permeability gap of the similar materials increases, and the permeability coefficient of the similar materials increases gradually. Figure 7a shows that the higher the calcium carbonate and gypsum content, the smaller the permeability coefficient, and that when the river sand–calcium carbonate–gypsum ratio is 50:3:7 and the paraffin wax content increases from 5% to 5.56%, the permeability coefficient decreases from $6.72 \times 10^{-6}$ cm/s to $1.87 \times 10^{-6}$ cm/s.

Figure 7b shows the effect of the calcium carbonate and gypsum content on the permeability coefficient of the similar materials. With the increase in the petroleum jelly–paraffin wax ratio, the material particles are filled with non-hydrophilic and viscous petroleum jelly; this can make the particles more compact and cohesive, and the resistance to water seepage increases obviously, and the permeability coefficient of the material decreases rapidly. Figure 7b also shows that the higher the auxiliary cementing agent content, the higher the petroleum jelly content, and the lower the permeability coefficient of the similar materials. When the petroleum jelly–paraffin wax ratio was 2:1, and the auxiliary cementing agent content increased from 15% to 22.22%, the permeability coefficient decreased from $8.34 \times 10^{-6}$ cm/s to $4.08 \times 10^{-6}$ cm/s.

The porosity and the content of cementing agent have significant effects on the permeability of the similar materials. Paraffin wax has the function of maintaining water in the sample, reducing dry cracks, and strengthening the compactness of the material. Petroleum jelly is an important non-hydrophilic cementing agent. It is proposed that the permeability coefficient of the material can be adjusted by changing the ratio of petroleum jelly–paraffin wax under a certain total mass of paraffin wax and petroleum jelly. At the same time, the content and proportion of the auxiliary cementing agent has considerable effects on the permeability coefficient. It can be concluded that the permeability coefficient of the aquifuge's similar materials is affected by the content of paraffin wax, petroleum jelly,

calcium carbonate, and gypsum. The expansion of the material is controlled by calcium carbonate and gypsum, which is helpful in the simulation of the secondary closure of the water contact layer after a crack is formed. The plasticity of the material is controlled by paraffin, which is helpful in the simulation of the large deformation of the water-resistant layer under low strength. The non-hydrophilicity of the material, which is controlled by Vaseline, helps to simulate the hydrologic properties of the water-resistant layer. While the content of calcium carbonate and gypsum controls the physical strength and hydrologic properties of the material, the permeability coefficient and the number can be inversely adjusted by changing the ratio of petroleum jelly–paraffin wax.

## 4. Solid–Liquid Coupling Simulation Test

The 8210 working face of the Majiliang Mine, located in Datong City, China, was the object of this experiment. The strike length of 8210 working face in the Majiliang Mine is 2533 m; the width of the working face is 200 m; the buried depth of the mining range is 450–550 m; the average thickness of the coal seam is 6.7 m; and the average dip angle of the coal seam is 1–4°. The fully mechanized mining method is complete with a roof collapse. Based on the research object and the experimental conditions, the geometric ratio of the model was determined to be 1:100. The experiment was conducted using the existing solid–liquid coupling model testing platform with the dimension of 3000 mm × 1600 mm × 200 mm in the laboratory. The research objective of this solid–fluid coupling similarity model experiment was to analyze the dynamic water filling and development of water filling channels in the 8210 working-face area caused by the aquifer overlying the roof of the working face.

Based on the geological and hydrogeological conditions of the Majiliang Mine, we established the statistical analysis of the specific physical strength and hydraulic parameters of the aquifuge (aquitard) layers. According to the solid–fluid coupling similarity theory (Section 2), the physical and mechanical strength and hydraulic parameters of similar materials for each rock layer were calculated. Following the above experiments, the conclusion of the experiment of the similar materials for the aquifuge (aquitard) layers was obtained, and the matching of the similar materials for each rock layer in the case of the aquifuge (aquitard) layers was determined. The ratio data of similar materials used are shown in Table 3.

After applying the similar model materials, the DIC digital speckle monitoring system [23] observation surface of the similar material was solidified and sealed using a molten-mixture ratio of paraffin wax–petroleum jelly of 1:1; after cooling and solidifying, the black speckle grid was sprayed on the model observation surface. To facilitate the monitoring of the DIC digital speckle monitoring system for the fluid–solid coupling similar model testing, a transparent plexiglass plate was fixed on the observation surface of the similar model, and a 5 mm thick lubricating silicone grease layer was uniformly smeared on the contact surface of the plexiglass plate and the similar material to seal the water flow. The target aquifer was then saturated with water, and the water tightness of each stratum and experimental device was checked at the same time. Finally, the physical model preparation before the solid–fluid coupling physical similarity model testing was completed.

**Table 3.** Statistical parameters of the similar materials ratio of each stratum.

| No. | Modeling Lithology | Thickness of Layer (m) | Uniaxial Compressive Strength (kPa) | Permeability Coefficient (×10⁻⁵ m/d) | Softening Coefficient | Ratio of Solid–Liquid Components | Ratio of River Sand–Calcium Carbonate–Gypsum | Ratio of Paraffin Wax–Petroleum Jelly |
|---|---|---|---|---|---|---|---|---|
| 15 | ② Coarse-grained sandstone | 10 | 139.23 | 4.7 | 0.55 | 9:01 | 30:03:07 | 1:01 |
| 14 | Sandy mudstone | 10 | 252.01 | 1.6 | 0.76 | 8:01 | 50:03:07 | 2:01 |
| 13 | Upper glutenite aquifer | 13.36 | — | 39.4 | — | — | — | — |
| 12 | Sandy mudstone | 10 | 252.01 | 1.6 | 0.76 | 8:01 | 50:03:07 | 2:01 |
| 11 | ② Coarse-grained sandstone | 20.23 | 139.23 | 4.7 | 0.55 | 9:01 | 30:03:07 | 1:01 |
| 10 | Sandy mudstone | 19.43 | 252.01 | 1.6 | 0.76 | 8:01 | 50:03:07 | 2:01 |
| 9 | ② Coarse-grained sandstone | 16.49 | 139.23 | 3.2 | 0.55 | 9:01 | 30:03:07 | 1:01 |
| 8 | Sandy mudstone | 5.66 | 252.01 | 1.6 | 0.76 | 8:01 | 50:03:07 | 2:01 |
| 7 | Lower sandstone aquifer | 4 | — | 9.7 | — | — | — | — |
| 6 | Siltstone | 4.99 | 170.37 | 6.8 | 0.81 | 8:01 | 50:03:07 | 1:02 |
| 5 | ① Coarse-grained sandstone | 3.19 | 193.16 | 6.8 | 0.25 | 8:01 | 40:03:07 | 1:02 |
| 4 | Fine sandstone | 5.29 | 211.76 | 6.8 | 0.83 | 8:01 | 40:03:07 | 1:02 |
| 3 | Glutenite | 8.14 | 178.16 | 5 | 0.75 | 8:01 | 50:03:07 | 1:02 |
| 2 | No.3 coal | 6.76 | 16 | 6.85 | — | — | 8:06:04 | — |
| 1 | Sandy mudstone | 22.46 | 252.01 | 1.6 | 0.76 | 8:01 | 50:03:07 | 2:01 |

NOTE: ① and ② respectively represent two types of coarse sand with different physical strength and hydraulic parameters, indicating that their similar material ratios are different.

The completed physical similarity model is shown in Figure 8.

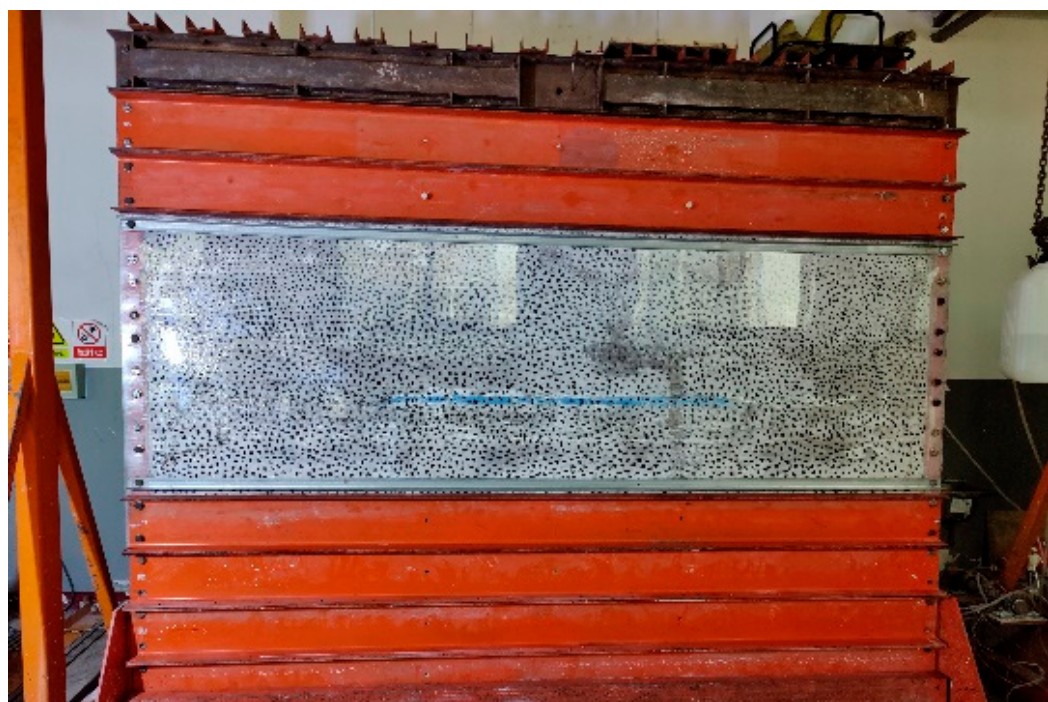

**Figure 8.** Physical similarity model for analyzing the water filling channel in the mining working face.

The visual monitoring system mentioned earlier for the dynamic development of the water filling channel in this experiment was the DIC digital speckle monitoring system, which is composed of two CCD digital cameras for image acquisition, with a camera resolution of 3000 × 4096 pixels, equipped with wide-angle lens, and installed on a rigid crossbar with a spacing of 1.04 m. The crossbar was fixed on a tripod, and the stereo-vision system was located approximately 2.3 m in front of the monitoring observation wall. Two sets of blue light-emitting diodes (LEDs) were positioned on the left and right fronts of the viewing wall to illuminate the monitoring surface, as shown in Figure 9.

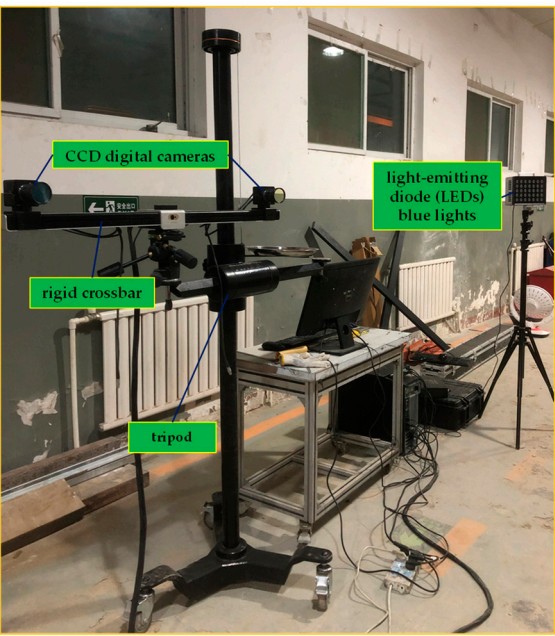

**Figure 9.** DIC digital speckle monitoring system.

The length of the simulated working face was 2000 mm; the boundary coal pillars were 500 mm and left on both sides of the model; the length of the open cut was 100 mm; the distance of the excavation was 100 mm; the excavation speed was 200 mm/h; and the mining height was 67.6 mm, gradually advancing from left to right, with a stopping line of 500 mm away from the right side. The initial pressure step of the working face was 500 mm, and the periodic pressure step was 300 mm. In the simulation test of solid–fluid coupling, it was found that the working face advances for a total of 2000 mm, and periodic pressure was applied 5 times.

The simulation results show that, when the working face was pushed 500 mm, the glutenite overlying the coal seam, in direct contact and with a thickness of 81.4 mm, as the direct top of the No. 3 coal seam, was fractured and collapsed for the first time, as shown in Figure 10a.

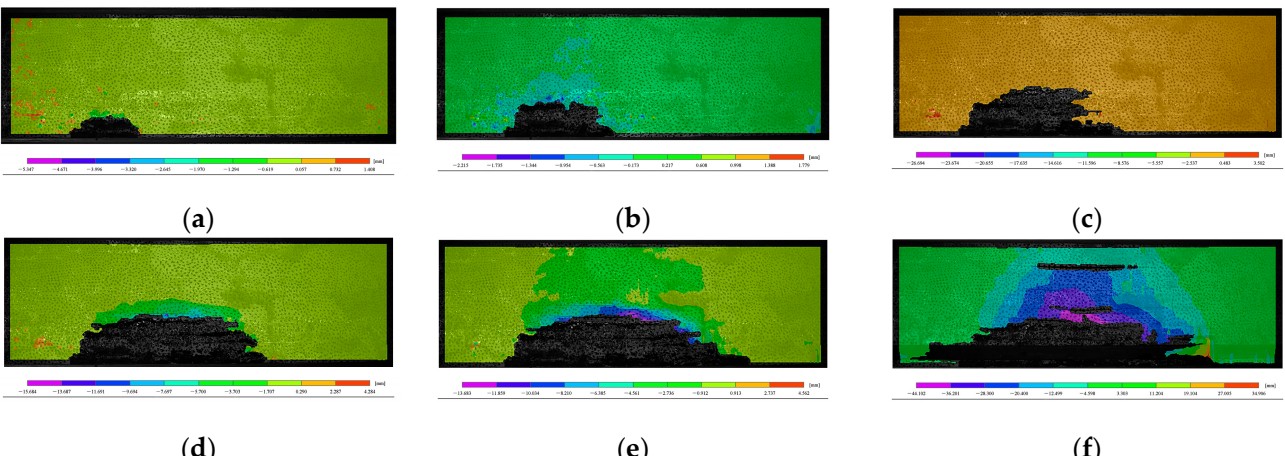

**Figure 10.** The displacement cloud diagram of the overlying rock strata of the coal seam during the advancement of the working face. (**a**) Advancing distance of the working face was 500 mm, directly breaking the roof for the first time. (**b**) Advancing distance of working face was 800 mm, and the main roof is broken for the first time. (**c**) Advancing distance of the working face was 1100 mm. (**d**) Advancing distance of the working face was 1400 mm. (**e**) Advancing distance of the working face was 1700 mm. (**f**) Advancing distance of the working face was 2000 mm.

When the working face advanced to 800 mm, the interbed of fine sandstone and coarse-grained rock, with a cumulative thickness in direct top contact of 134.7 mm, as the main roof of the No. 3 coal seam, broke and collapsed during the second weighting of the working face, as shown in Figure 10b. The coal seam roof collapsed during the second pressure, causing roof cracks to develop at the bottom of the weak aquifer (the lower aquifer) close to the coal, resulting in a serious roof cutting and collapse. At the same time, the amount of water flowing into the working face slowly increases, mainly from the weak water rich aquifer in the coal seam collapse zone.

As the working face continued to advance to 1100 mm, 1400 mm, 1700 mm, and 2000 mm, the coal seam was overburdened, developing a failure and movement pattern, as shown in Figure 10c–f. In addition to the progressive collapse of the direct roof, the main roof also broke and collapsed periodically, and the judging periodic weighting and breaking distance was 300 mm. The movement of overlying strata was not completely stable when the mining line was pushed to 2000 mm. The goaf space of the working face was closed after the mining of the No. 3 coal seam was stopped, and the water-filling condition of the upper and lower aquifers was maintained; the monitoring of the stability movement of overlying strata in the goaf continued.

The monitoring revealed that the separated layer fractures of the lower aquifer and the upper strata continued to extend transversely and were affected by the dilatancy of the caving block in the caving zone. Meanwhile, the low aquifer and the overlying aquifuge

(aquitard) strata gradually produced a local development along the joint surface of each stratum. Along with the progressive compaction of the lower separated layer fractures, the stratification phenomenon continued to develop upwards. At the same time, vertical fractures developed in the shear fracture form of each rock stratum, appearing in the pinching out area of the separated fractures. The separated layer fractures and vertical fractures of the overlying strata developed and were alternated in time and space, forming the main water-filling channel of the underlying working face.

After 7.8 h after the goaf was closed, the movement of the overlying rock with a separated layer fissure was observed, and a stable state diagram is shown in Figure 11. This state was maintained until the end of the monitoring observation experiment. In this process, the horizontal and vertical fractures of the overlying strata in the goaf gradually developed upwards to the upper sandstone aquifer. Meantime, the water filling flow rate of the working face did not change much, but the water-filling duration was relatively long. At the same time, under the action of fluid–solid coupling, the similar materials in the rock stratum begun to expand and extrude in the direction of the cracks in the water-filling channel. Compared with the vertical fractures, the fracture space of the horizontal separation layer was more affected by compression. However, the growth rate of vertical fractures was weakened because the water-carrying sand in the upper aquifer filled and closed the vertical fractures, and the overall permeability coefficient of the water-filling passageway was reduced. The water-filling effect weakened, so that the working face of the water flowed until disappearing.

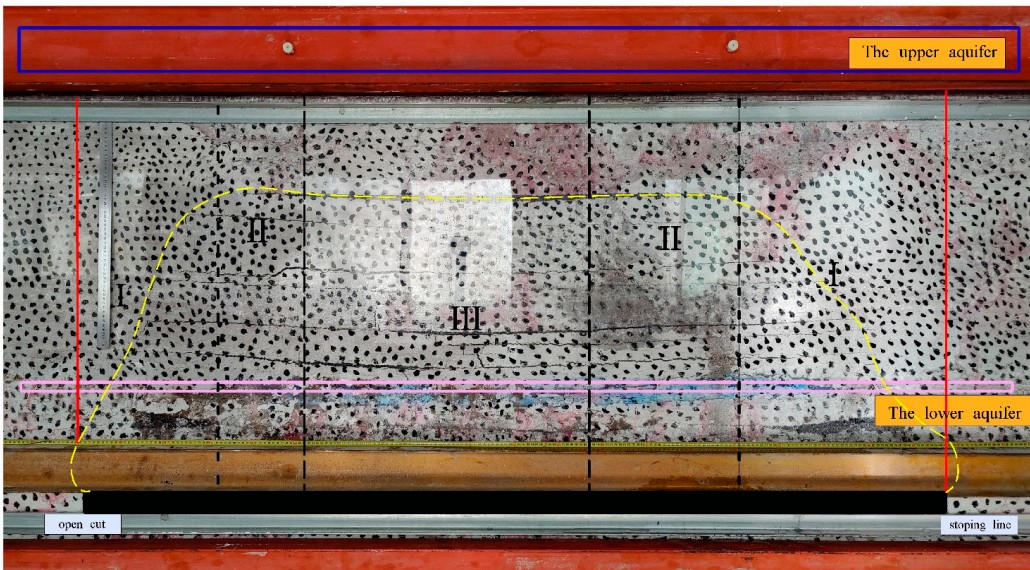

**Figure 11.** Distribution diagram of water-conducting cracks in the overlying strata of the goaf under fluid–solid interaction. I. The development area of terraced vertical fissures on the side of coal pillar. II. The zone of delamination fracture development. III. Large overhanging tension vertical fracture development area.

It was observed that the dynamic development of the water-filling passageway in the goaf is essentially the dynamic development process of the separated layer fractures and vertical fractures in the overlying strata of the goaf in the working face. Among them, the development of the separation layer fractures drives the progressive development of the vertical main water filling fractures. Under the fluid–solid coupling effect, similar materials in the aquiclude (aquitard) have the effect of water absorption expansion and weakening of physical strength. At the same time, it plays a major role in controlling the evolution of water-filling channels in the goaf. It was found that the developed range of the water-conducting fissures is similar to a trapezoid, the interface length of trapezoid roof is 121 m, and the normal height of failure is 90 m. The development of the overlying rock

fissures in the goaf under the fluid–solid coupling action is divided into water-conducting fissures, as shown in Figure 11.

As shown in Figure 11, the experimental results show that the development of cracks and water gushing can be divided into the following three areas. I. The development area of the terraced vertical fissures on the side of coal pillar, in the range of 0~30 m in front of the coal wall. The strata above the coal seam are affected by the advance abutment pressure of the mining face under the action of high-stress superposition; the crack initiation condition is gradually satisfied (the tensile strain reaches the ultimate tensile strain of the rock and is related to the rotation angle of the rock beam). The crack tip begins to expand, the crack opening degree gradually increases, and the rock stratum fissures develop in this area, but not entirely. II. The zone of delamination fracture development, in the range of the initial weighting step and before the last weighting cycle. The key blocks of the overburdened rock slide and lose stability, and the areas of tensile failure and shear failure of each rock stratum run through to the position of the direct top of coal seam, above the top of the goaf. There are many separate layer fissures formed by uneven subsidence, and there is separate layer mixing due to the difference of rock mass thickness and strength. In this range, the roof delamination fractures are unstable structures, which lead to tensile fracture and perforated fracture in further rock migration and become a good channel to conduct the upper glutenite aquifer. III. Large overhanging tension vertical fracture development area. In the middle region of the II zone, the roof separation fracture is an unstable structure, and tensile fracture occurs in further rock migration and forms a through fracture, thus being a good channel to conduct the upper glutenite aquifer.

This indicates that fluid–solid coupling and physical similarity simulation experiments are feasible ways to explore the controlling factors of water-filling in the goaf during mining.

## 5. Conclusions

In this paper, a new type of aquifuge simulation material was developed and experimentally investigated regarding physical–mechanical strength and hydraulic parameters. It was applied to a similar model experiment of fluid–solid coupling. Based on the results and analysis, the following conclusions were obtained:

(1) Through a large number of proportioning experiments, a mixture of river sand, calcium carbonate powder, gypsum, paraffin wax, and petroleum jelly was developed. The fluid–solid coupling of similar materials can satisfy the similarity requirements of model test strength, water physics, and other types of aquifuge (aquitard).

(2) The mechanical parameters and permeability coefficient of the material can be adjusted within a wide range: the range of the compressive strength is 16.99–426.47 kPa, the range of the softening coefficient is 0.660–0.805, and the range of the permeability coefficient is $1.01 \times 10^{-7}$–$8.34 \times 10^{-7}$ cm/s. Therefore, this type of similar material can simulate various low- and medium-strength aquifuge (aquitard) rock mass materials with different permeabilities.

(3) The permeability coefficient of the similar materials is affected by the content of paraffin wax, petroleum jelly, calcium carbonate powder, and gypsum. While the content of calcium carbonate and gypsum mainly controls the physical strength and hydrologic properties of the material, the permeability coefficient can be adjusted inversely by changing the ratio of paraffin wax–petroleum jelly. The expansion is controlled by calcium carbonate and gypsum, which is helpful in the simulation of the secondary closure of the water contact layer after crack formation, and the plasticity is controlled by paraffin, which is helpful in the simulation of the large deformation of the water contact layer under a low strength. The non-hydrophilicity is controlled by petroleum jelly, which helps to simulate the hydrologic properties of the aquifuge (aquitard).

(4) This type of similar material was successfully applied in the fluid–solid coupling model testing of the coal seam excavation face in the Majiliang Coal Mine, Datong City, China; the material guarantee was provided for the successful results of the

experiment. In this fluid–solid interaction similar model test, it was found that, under the condition of double water-filled aquifers in the overlying strata of the goaf, the evolution law of the overlying strata water channel in the goaf is phased, especially after the goaf is closed, when the upper aquifer is not in the range of the water-conducting fissure zone in the closure stage of the mined-out area. However, when the distance is very close, the existence of the water-conducting fissure zone has a considerable influence on the continuous stable movement of the overlying rock, and the water-filled channels formed by the fluid–solid interaction can be divided into three symmetrical regions: I. the development area of terraced vertical fissures on the side of coal pillar; II. the zone of delamination fracture development; and III. a large overhanging tension vertical fracture development area.

**Author Contributions:** Conceptualization, X.S.; investigation, X.S. and D.Z.; methodology, X.S., Q.L., D.Z. and G.F.; software, X.S.; validation, X.S.; formal analysis, X.S. and J.L; resources, G.F. and J.L.; data curation, X.S.; writing—original draft preparation, X.S.; writing—review and editing, X.S.; visualization, X.S.; supervision, G.F. and J.L.; project administration, G.F. and D.Z.; funding acquisition, X.S., G.F. and J.L. All authors have read and agreed to the published version of the manuscript.

**Funding:** This research was funded by the Distinguished Youth Funds of National Natural Science Foundation of China (51925402), Tencent Foundation or XPLORER PRIZE, and Shanxi Province Key Laboratory Construction Project Funds (202104010910021).

**Institutional Review Board Statement:** Not applicable.

**Informed Consent Statement:** Not applicable.

**Data Availability Statement:** Not applicable.

**Conflicts of Interest:** The authors declare no conflict of interest.

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
