# Peer review of "Experimental Study on the Properties of Simulation Materials for an Aquifuge for a Fluid–Solid Coupling Physical Similarity Model Test"

_applsci, doi:10.3390/app13158667_

Round 1

Reviewer 1 Report

The problem of water inrush from coal mines has become increasingly prominent in coal mining in recent years. Once the water filling in the goaf reaches a certain water pressure threshold, the water in the goaf will break through the waterproof rock below the goaf floor and enter the underlying coal face. This situation not only seriously threatens the safety of underground miners, but also will cause huge economic losses to the coal mining industry. Therefore, the development and proportion research of similar materials for aquifuge (aquitard) layers are the foundation and guarantee of model testing. Herein, Shen et al. developed a new type of aquifuge simulation materials, which used river-sand as the skeleton of the material, gypsum and calcium-carbonate powder as auxiliary cementing-agent, paraffin-wax and petroleum-jelly as waterproof cementing-agent. Through orthogonal test and systematic analysis, the influence mechanism of different proportions of raw materials on the variation of physical mechanical strength and hydraulic parameters was studied. The reported results are useful to coal mine engineering and of fair significance. I suggest a minor revision.

(1)    The authors concluded that a new type of aquifuge simulation materials was developed and experimentally investigated on the physical mechanical strength and hydraulic parameters, finally, it was applied to a similar model experiment of fluid-solid coupling. What is more persuasive is that a comparison with previous aquifuge simulation materials is made and discussed.

(2)    The authors stated that the proposed type of similar material has been successfully applied in the fluid-solid coupling model test of the coal seam excavation face in Majiliang coal mine, Datong City, China, the material guarantee is provided for the good result of the experiment. Could the authors provide some engineering application data to support this conclusion?

(3)    The unit of uniaxial compression strength should be “kPa”, but not “KPa”. Please revise it at main text of the manuscript and in Table 1 and Figures 3 and 4.

(4)    There are many minor errors or typos in the main text of the manuscript and a careful proofreading is essential to correct these errors. <i> Line 318, “From figure 4” should be corrected to “From Figure 8”. <ii> Lines 15-17, “In order to meet … on the mine water inrush. A new type of …” should be corrected to “In order to meet … on the mine water inrush, a new type of …”. <iii> Line 247, “in figure 2” should be corrected to “in Figure 2”. <iii> Lines 377, 384, 389, “figures 5.(a) and 5.(b)” should be corrected to “Figures 5a and 5b”. <iv> Line 403, “Where KR is the…” should be corrected to “where KR is the”. <v> Line 405, “in table 2 and figure 6” should be corrected to “in Table 2 and Figure 6”. <vi> Lines 580-581, “. As shown in Figure 9. (b)- 9. (d)” should be corrected to “, as shown in Figures 9b-d”. <vii> And so forth.

(5)    There are format errors in the Reference List. <i> In Ref. [1], “New Type of Similar Material for Simulating the Processes of Water Inrush from Roof Bed Separation. ACS OMEGA” should be corrected to “New type of similar material for simulating the processes of water inrush from roof bed separation. ACS Omega”. <ii> In Ref. [9], the article title “New Similar Material Simulation Test of Overburden Rock’s Separation” should be corrected to “New similar material simulation test of overburden rock’s separation”. <iii> In Ref. [18], the journal name “Journal of Liaoning technical university(natural science)” should be corrected to “Journal of Liaoning Technical University(Natural Science Ed.)”. <iv> In Ref. [20], “Experimental Study on the Basic Properties of a Green New Coal Mine Grouting Reinforcement Material. ACS OMEGA” should be corrected to “Experimental study on the basic properties of a green new coal mine grouting reinforcement material. ACS Omega”. <v> In Ref. [21], the journal name “B ENG GEOL ENVIRON” should be corrected to “B. Eng. Geol. Environ.”.

It is essential to carefully revise English sentences through all parts of the manuscript.

Author Response

Thank you for taking the time to review the content of my article. Your recognition of my work is the biggest motivation for me to move forward. Thanks again for your professional commentary and rigorous review style. Reply to your audit comments with details in the attachment file and re-edit the manuscript attachment

Reviewer 2 Report

Dear Dr.,

The article with the title “Experimental study on properties of a simulation materials of aquifuge for fluid-solid coupling physical similarity model test” is suitable for Geomechanics. But the manuscript is required to be modified. The paper can be accepted if the following comments are described. Please explain and add them to the manuscript with the red color.

1)     Some grammatical errors are seen in the article. The English language of the manuscript should be edited by a native English speaker.

2)     The title is required to be modified. I think it is not a fluid-solid coupling, it is only a mixing of fluid and solid. Could you explain what is your purpose for suggesting this keyword namely coupling?

3)     The abstract doesn't have novelty in it. The authors should rewrite the abstract with the main novelty in it. I think this paper is similar to a technical report and it does not have any new goal.

4)     In the introduction, the introduction should be improved to show the difference between this study relative to the previous ones. What is the main purpose of the article? The difference between your paper with the previous papers is not clear. What the advantages of your paper is?

5)     Keywords: why do some words start with a capital letter and others start with the small alphabet?

Section 2: I don’t think this section can present the fluid-solid coupling, especially equation 1. I suggest you read and cite these papers as a good scheme to make a coupling between fluid and solid (Hydromechanical coupling or coupling of reservoirs and geomechanics), especially the fundamental equations. The papers are:

Duran, O., Sanei, M., Devloo, P. Santos, E. 2020. An enhanced sequential fully implicit scheme for reservoir geomechanics. Journal of Computational Geosciences. https://doi.org/10.1007/s10596-020-09965-2.

Sanei, M., Duran, O., Devloo, P. Santos, E. 2021. Analysis of pore collapse and shear-enhanced compaction in hydrocarbon reservoirs using coupled poro-elastoplasticity and permeability. Arabian Journal of Geosciences. https://doi.org/10.1007/s12517-021-06754-8.

Sanei, M., Duran, O., Devloo, P. Santos, E. 2022. Evaluation of the impact of strain-dependent permeability on reservoir productivity using iterative coupled reservoir geomechanical modeling. Geomechanics and Geophysics for Geo-Energy and Geo-Resources. https://doi.org/10.1007/s40948-022-00344-y.

Sanei, M., Ramezanzadeh, A. 2022. Building 1D and 3D static reservoir geomechanical properties models in the oil field. Journal of Petroleum Exploration and Production Technology. https://doi.org/10.1007/s13202-022-01553-7.

6)     Figure 2: How this figure can show the fluid-solid coupling?

7)     Figure 2 and Figure 3 are shown a clear result. Please let me know what the novelty of this section comparing with previous research. Explain it in the manuscript.

8)     Figure 4 and Figure 5 are shown clear results. Please let me know what is the novelty of these figures.

9)      Section 3.4.2: How the permeability test was done? What is the goal of equation 4? How you can use these equations 4 and 5 in the permeability test?

10) Section 4: I don’t understand the meaning of simulation. Could you explain how the simulation was done? What was the numerical modeling for this simulation? Could explain the method and the condition of modeling?

11) Conclusions lack of novelty. Please rewrite your conclusions. What were your goals?  It is not clear to the reader why you have done these tests. What was the clear conclusion?

Best regards,

It is required to be modified.

Author Response

(The authors gave the same response as above.)

Reviewer 3 Report

Review

Title: Experimental study on properties of a simulation materials of aquifuge for fluid-solid coupling physical similarity model test

Commentary: In order to study the possibility of fulfilling the special requirements of physical-mechanical strength and high water resistance of water-bearing aquifer materials in fluid-solid coupling studies, the Authors developed a physical simulation model for the study of mine water inflow. Equivalent materials were prepared for model-simulation studies for the water-bearing layer.

The equivalent material of the water-bearing layer was produced according to the specified proportion of components and the established technological process.

Using the orthogonal test and systematic analysis, the mechanism of the effect of different proportions of components on changes in physical mechanical strength and hydraulic parameters was studied.

The work is experimental in nature, and the results are interesting.

Reviewer's decision: The article can be published in a journal after correcting errors, mainly editorial. All noticed defects are presented in *.pdf file.

The article is written in understandable English. It does not contain blatant errors of lexical language.

Author Response

(The authors gave the same response as above.)
